# Evaluation of the Sensitivity of SMOS L-VOD to Forest Above-Ground Biomass at Global Scale

**Arnaud Mialon [1],* , Nemesio J. Rodríguez-Fernández [1] , Maurizio Santoro [2] , Sassan Saatchi [3], Stéphane Mermoz [1,4] , Emma Bousquet [1] and Yann H. Kerr [1]**

1   CESBIO Centre d'Etudes Spatiales de la Biosphère Université de Toulouse, CNES/CNRS/INRAE/IRD/UPS, 18 av. Edouard Belin, bpi 2801, 31401 CEDEX 9 Toulouse, France; nemesio.rodriguez@cesbio.cnes.fr (N.J.R.-F.); mermoz@globeo.net (S.M.); emma.bousquet@cesbio.cnes.fr (E.B.); yann.kerr@cesbio.cnes.fr (Y.H.K.)
2   Gamma Remote Sensing, Worbstrasse 225, 3073 Gumligen, Switzerland; santoro@gamma-rs.ch
3   Jet Propulsion Laboratory, California Institute of Technology, Pasadena, CA 91109, USA; saatchi@jpl.nasa.gov
4   Globeo, (Global Earth Observation), Avenue Saint-Exupery, 31400 Toulouse, France
*   Correspondence: arnaud.mialon@cesbio.cnes.fr; Tel.: +33-561558524

**Abstract:** The present study evaluates the L band Vegetation Optical Depth (L-VOD) derived from the Soil Moisture and Ocean Salinity (SMOS) satellite to monitor Above Ground Biomass (AGB) at a global scale. Although SMOS L-VOD has been shown to be a good proxy for AGB in Africa and Tropics, little is known about this relationship at large scale. In this study, we further examine this relationship at a global scale using the latest AGB maps from Saatchi et al. and GlobBiomass computed using data acquired during the SMOS period. We show that at a global scale the L-VOD from SMOS is well-correlated with the AGB estimates from Saatchi et al. and GlobBiomass with the Pearson's correlation coefficients (R) of 0.91 and 0.94 respectively. Although AGB estimates in Africa and the Tropics are well-captured by SMOS L-VOD (R > 0.9), the relationship is less straightforward for the dense forests over the northern latitudes (R = 0.32 and 0.69 with Saatchi et al. and GlobBiomass respectively). This paper gives strong evidence in support of the sensitivity of SMOS L-VOD to AGB estimates at a globale scale, providing an interesting alternative and complement to exisiting sensors for monitoring biomass evolution. These findings can further facilitate research on biomass now that SMOS is providing more than 10 years of data.

**Keywords:** L-VOD; AGB (Above Ground Biomass); SMOS (Soil Moisture and Ocean Salinity)

## 1. Introduction

Above Ground Biomass (AGB), while a key component of the Earth's Surface, plays a major role in the carbon cycle [1–3] and is considered an essential climate variable. In the context of climate change, it is essential to better understand the role of AGB [4–7] and its interactions with the atmosphere [3,8] to better estimate the carbon stocks. It is therefore essential to monitor its evolution globally on a timely scale.

In this respect, satellite remote sensing is an essential tool as it provides observations for the entire Earth surface with regular revisits depending on the characteristics of the sensor. Various static maps of AGB obtained by combining datasets acquired at different wavelengths (from optical to radar), exist [9–12]. Monitoring dense biomass is challenging and hence existing datasets sometimes differ on their AGB estimates [13]. This is particularly true for dense tropical forests where optical and radar measurements tend to saturate [10,14]. Future missions such as BIOMASS [15] and NASA-ISRO

Synthetic Aperture Radar (NISAR) [16] are planned to provide more accurate spatial and temporal estimates of the biomass. In the meantime, a particular attention has been recently put on the use of the passive microwaves to monitor AGB even though passive microwave sensors are characterized by low spatial resolutions. Recent studies highlighted a strong relationship between the Vegetation Optical Depth (VOD) and AGB estimates [17,18]. Rodríguez-Fernández et al. [19] showed a stronger sensitivity of L-band VOD to AGB compared to higher frequencies (X and C bands) and also lower saturation for L-band at high AGB values. In addition, the evolution of the carbon stock estimation of the dryland Savannahs during the SMOS period (2010–2017) was evaluated in Africa [20] and for the tropics [21]. While these studies mainly focused on the Tropics and Africa, the L-VOD has not been evaluated in boreal and temperate forests, despite the importance of Northern latitudes to the carbon budget [22]. Yet, Northern latitudes represent a widespread forest biomes on Earth, with a significant impact on the climate system [4,5,7]. The amount of carbon stocked in boreal and temperate forests represents approximately 27% of the total amount of forest carbon [23].

The purpose of this paper is thus to evaluate the sensitivity of L-VOD to AGB with respect to two recent AGB datasets of reference at global scale, including the northern latitudes.

## 2. Data and Methods

### 2.1. Data

#### 2.1.1. SMOS and L-VOD

SMOS is a satellite mission operated by ESA (European Space Agency) and CNES (Centre National d'Etudes Spatial, French space agency) launched in November 2009 [24]. It is equipped with an L-band 2-D interferometer that measures the emission of the Earth surface at 1.4 GHz (L-band). The main objective of the SMOS mission over land is to derive daily surface soil moisture at global scale [25] with a complete coverage of the Earth surface within 3 days. Thanks to the multi-angular brightness temperatures (TB) of SMOS, both surface soil moisture and vegetation optical depth are derived [26] by using the radiative transfer model L-Meb (L-band Microwave Emission of the Biosphere, Wigneron et al. [27]). The vegetation layer contributes to the radiative emission at L-band by scattering the emission from the underlying surface and by emitting its own radiation. These contributions are taken into account in the parameter $\tau$ of the L-Meb model [27,28] also known as the L-VOD. The L-VOD used here is obtained from SMOS-IC (Inra Cesbio) product [29] as it was shown to exhibit stronger relationships with AGB estimates than other SMOS datasets [19–21]. SMOS-IC dataset is projected on the EASE (Equal Area Scalable Earth) Grid version 2 [30] with a spatial resolution of 25 km $\times$ 25 km at 30° of latitude.

#### 2.1.2. AGB Saatchi et al.

The AGB estimates by Saatchi et al. [12], updated in Reference [21], were derived as follows. The method was calibrated using in-situ plots that spanned a variety of forest types on Latin America, Subsaharan Africa and Southeast Asia, with a minimum size of 0.1 ha and the biomass densities representing all trees >10 cm in diameter. To compensate for the lack of systematic spatial sampling of AGB from in-situ plots, a method for estimating AGB from the satellite GLAS (Geoscience Laser Altimeter System) LiDAR (light detection and ranging) measurements of forest vertical structure was included. The spatial estimates of AGB (i.e., the AGB map) were then derived by combining satellite images and using a maximum entropy (MaxEnt) approach calibrated using the AGB estimates defined above. The remote sensing data included products from different earth observing optical (MODIS, Landsat) and radar (QSCAT, ALOS2, SRTM) sensors, to derive metrics sensitive to vegetation cover, density, seasonality, moisture, roughness, and surface topography. The Saatchi global map (Figure 1) has a spatial resolution of 30 m.

### 2.1.3. GlobBiomass

To generate a dataset of AGB at high resolution for all Earth's land surfaces, Santoro et al. combined a large pool of spaceborne remote sensing observations from two synthetic aperture radar (SAR) missions (Envisat and ALOS) and used optical (Landsat) and LiDAR (Icesat GLAS) data in support of the model calibration procedure. The observations of the SAR backscatter were channeled into two separate retrieval algorithms, each providing an estimate of the growing stock volume (GSV). that is, the density of the tree volume per unit area. To compensate for sensor-specific retrieval errors, the GSV estimates were combined to form a final value of GSV. AGB (middle Figure 1) was obtained from the GSV using spatially explicit estimates of wood density and stem-to-total biomass expansion factors [31–33]. The AGB and GSV datasets, referred to as GlobBiomass data products, have a spatial resolution of 100 m and are representative of the year 2010. A validation at the hectare scale showed that the GlobBiomass dataset captures the spatial variability of AGB globally. Nonetheless, the estimates are affected by an increasing underestimation in wet tropical and temperate forests when AGB is above 250 Mg/ha.

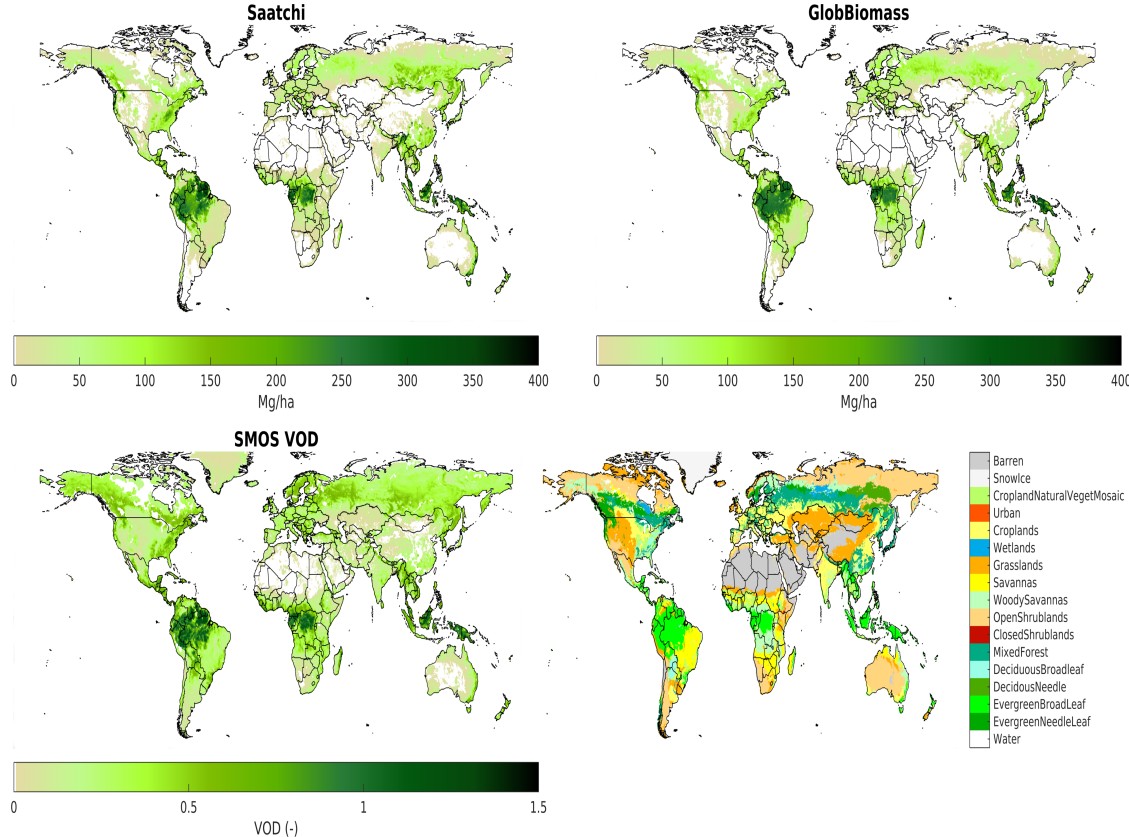

**Figure 1.** Above ground biomass (AGB) global maps (Saatchi **top left**, GlobBiomass **top right**) and Soil Moisture and Ocean Salinity (SMOS) derived L band Vegetation Optical Depth (L-VOD) map for 2015 (**bottom left**). **Bottom right** panel shows the International Geosphere-Biosphere Program (IGBP) land surface classification.

### 2.1.4. Land Surface Classification IGBP

International Geosphere-Biosphere Program (IGBP) land surface classification [34] identifies 17 ecosystems (Figure 1). The initial map is resampled on the EASE grid version 2 by allocating to each pixel the dominant class. Only pixels with a dominant fraction of single IGBP class higher than 80% are considered. Pixels characterized by a fraction of open water body larger than 5% or by a fraction of ice/snow larger than 10% are excluded from the analysis as these conditions impact significantly

SMOS observations and so the quality of the SM/VOD retrieval. This leads to 56% of the pixels over land selected for our analyses (i.e., 115,004 pixels out of 206,256).

**Table 1.** IGBP land cover classification, grouped into aggregated classes

| IGPB | Aggregated Classes |
|---|:---:|
| Evergreen Needleleaf and Broadleaf Forests<br>Deciduous Needleleaf and Broadleaf Forests, mixed Forests | Dense vegetation, forest |
| Closed and open Shrublands, Woody Savannahs, Savannahs<br>Grasslands, wetlands, barren, Cropland and Natural Vegetation Mosaics | low vegetation |
| Urban and Built-Up, Snow Ice, water bodies | Not considered |

*2.2. Methodology*

First, the SMOS data are filtered because the quality of the retrievals can be affected by the presence of Radio Frequency Interference (RFI) [35]. The RMSE between the modeled TB (obtained with the L-Meb model) and SMOS measured TB is used as an indicator of the retrieval quality [21] and only L-VOD with a RMSE < 8 K are kept. Strong topography is excluded as it significantly impacts SMOS TB angular signature. Then the L-VOD are averaged over a year using both ascending and descending orbits. This removes the diurnal variations of L-VOD due to its sensitivity to vegetation water content and rain interception. All SMOS years were tested without significant differences. For SMOS, 2010 is not optimal because of the commissioning phase so 2015 is chosen as it corresponds with the update of Saatchi's AGB dataset. Second, the Saatchi and the GlobBiomass high resolution AGB maps are projected onto the SMOS coarse resolution grid, by averaging high resolution pixels fitting into each low resolution pixel. Finally, SMOS pixels are splitted into 2 main groups one for forest classes and one for low vegetation classes (Table 1). The sensitivity of L-VOD to AGB estimates is discussed for the forest and the low vegetation groups classes, by using the Pearson's correlation coefficient (R). The discussion focuses at global scale and more specifically for the northern latitudes (i.e., latitudes > 35°N ) and the tropics (latitudes < 25°N and > −25°S). The details per IGBP classes are presented in the Supplementary Materials.

## 3. Results

*3.1. Analysis at Global Scale*

The relationships of L-VOD vs AGB are presented in Figure 2 for dense forests, low vegetation and a unified class. Considering all classes, the L-VOD vs AGB is well represented by a logistic function as in Rodríguez-Fernández et al. [19]. The two fits (with Saatchi and GlobBiomass Figure 2) are similar except around L-VOD values of 0.9 where Saatchi AGB are slightly lower than GlobBiomass AGB with values around 220 Mg/ha and 250 Mg/ha respectively. On one hand, the distribution is less spread for GlobBiomass for AGB below 150 Mg/ha and L-VOD < 0.7. On the other hand, the distribution of GlobBiomass tends to saturate for AGB values higher than 250 Mg/ha. Overall, the agreement between L-VOD and the two AGB maps is similar as shown by the correlation coefficients R = 0.91 for Saatchi and R = 0.93 for GlobBiomass (Table 2). The dense forest case is also characterized by a logistic function, but the relationship between L-VOD and AGB presents a slight difference, with GlobBiomass being more correlated to L-VOD (R = 0.84) than Saatchi (R = 0.73). Finally, the L-VOD shows a similar correlation (R = 0.75) with GlobBiomass and Saatchi for low vegetation classes.

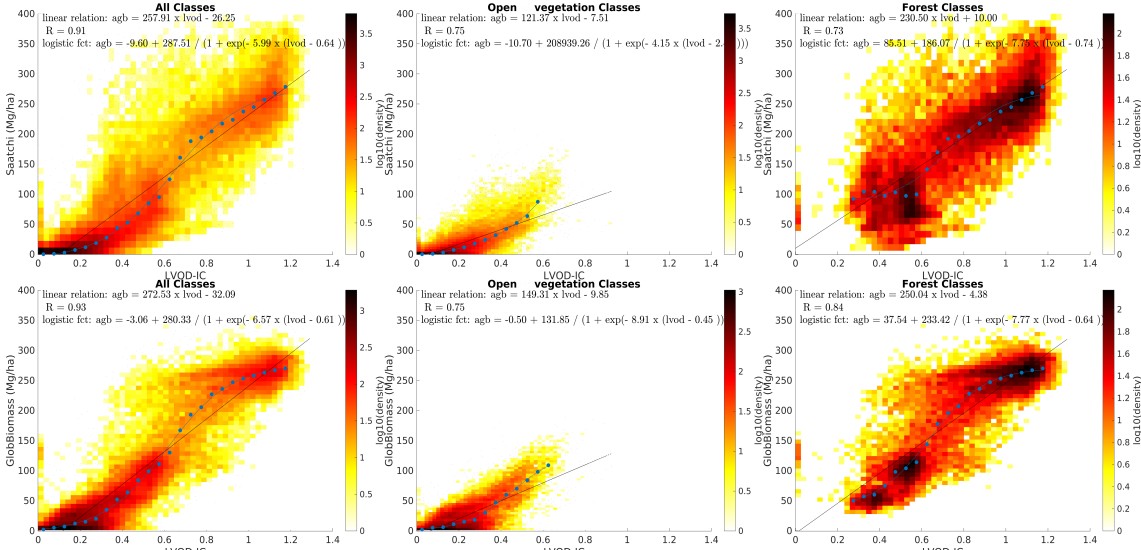

**Figure 2.** AGB as a function of L-VOD at global scale, using Saatchi (**top row**) and GlobBiomass (**bottom row**), considering all the IGBP classes (**left column**), the IGBP classes of the low vegetation group (**middle column**) and the IGBP classes of the forest group (**right column**). See Table 1 for the definition of the groups.

**Table 2.** Comparison between L-VOD and AGB maps from Saatchi and GlobBiomass at global scale; the statistics reported are the correlation coefficient (R), considering (i) all the IGBP classes (ii) the forest classes (iii) the low vegetation classes.

| Region | Global | | High Lat. | | Tropics | |
|---|---|---|---|---|---|---|
| **AGB** | **R** | **nb pt** | **R** | **nb pt** | **R** | **nb pt** |
| | | | All Classes | | | |
| Saatchi | 0.91 | 76,305 | 0.76 | 27,308 | 0.92 | 76,305 |
| GlobBiomass | 0.94 | 60,041 | 0.85 | 21,472 | 0.94 | 60,041 |
| | | | Forest Classes | | | |
| Saatchi | 0.73 | 21,119 | 0.32 | 5741 | 0.62 | 21,119 |
| GlobBiomass | 0.84 | 21,120 | 0.69 | 5741 | 0.67 | 21,120 |
| | | | Low veget. Classes | | | |
| Saatchi | 0.76 | 55,186 | 0.66 | 21,567 | 0.80 | 55,186 |
| GlobBiomass | 0.75 | 38,921 | 0.58 | 15,731 | 0.85 | 38,921 |

## 3.2. Northern Latitudes

The case of northern latitudes is presented in Figure 3. Compared to tropical forests where AGB estimates reach 500 Mg/ha and L-VOD reaches 1.2, the sparse canopy of the northern forests implies a smaller AGB range, that is, 0 to 250 Mg/ha and L-VOD below 0.8 (Figure 3). The L-VOD is more correlated to GlobBiomass (R = 0.83) than to Saatchi (R = 0.76, Table 2) when considering all the classes (left column Figure 3), whereas the L-VOD presents the same dispersion with the 2 AGB maps. The difference of correlation R is mainly caused by the forest group (Figure 3) for which L-VOD is not well correlated to Saatchi (R = 0.32 when compared to 0.7 with GlobBiomass).

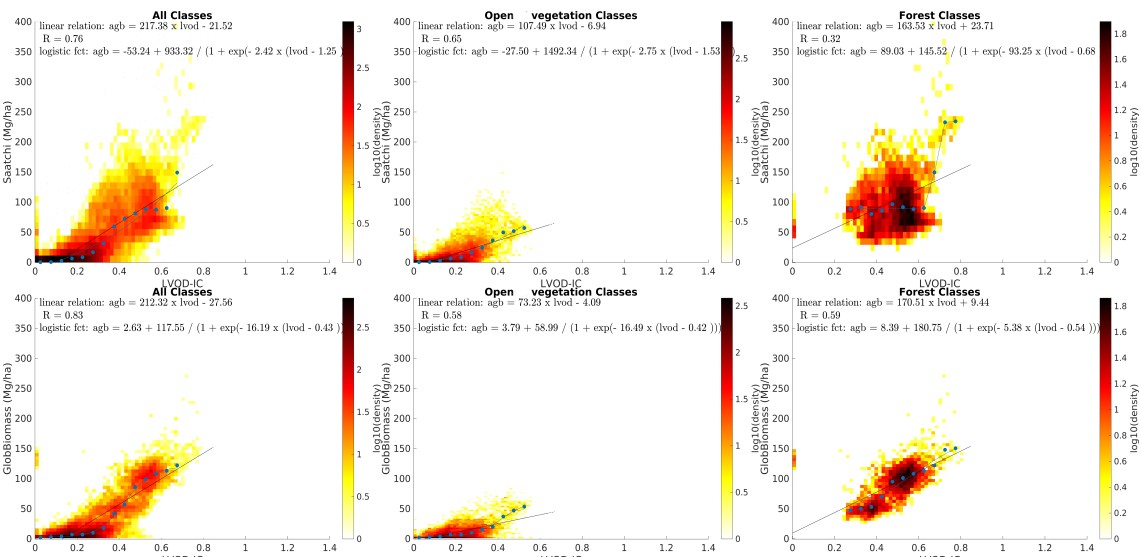

**Figure 3.** AGB Saatchi (**top row**) and GlobBiomass (**bottom row**) as a function of L-VOD over the northern high latitudes. **left column** Figures are for all IGBP classes, **middle column** for low vegetation classes, and **right column** are for forest classes.

### 3.3. Tropics

The comparisons between L-VOD and AGB for the tropics are presented in Figure 4 and Table 2. The relationships follow a sigmoid shape (Figure 4) as reported in Rodríguez-Fernández et al. [19], with correlation coefficients R equivalent for Saatchi and GlobBiomass (0.92 and 0.94 respectively, Table 2). Nevertheless, the AGB estimates from the GlobBiomass dataset tend to cluster between 200 and 300 Mg/ha at around 250 Mg/ha for L-VOD > 0.7 (Figure 4). In the panel of the forest class, we identified a separate cluster for L-VOD values between 0.4 and 0.8 and an AGB around 50 Mg/ha for GlobBiomass and between 0 and 50 Mg/ha for Saatchi (Figure 4) with lower AGB values than the rest of the distributions. These pixels correspond to the deciduous broadleaf class that is discussed in the Supplement.

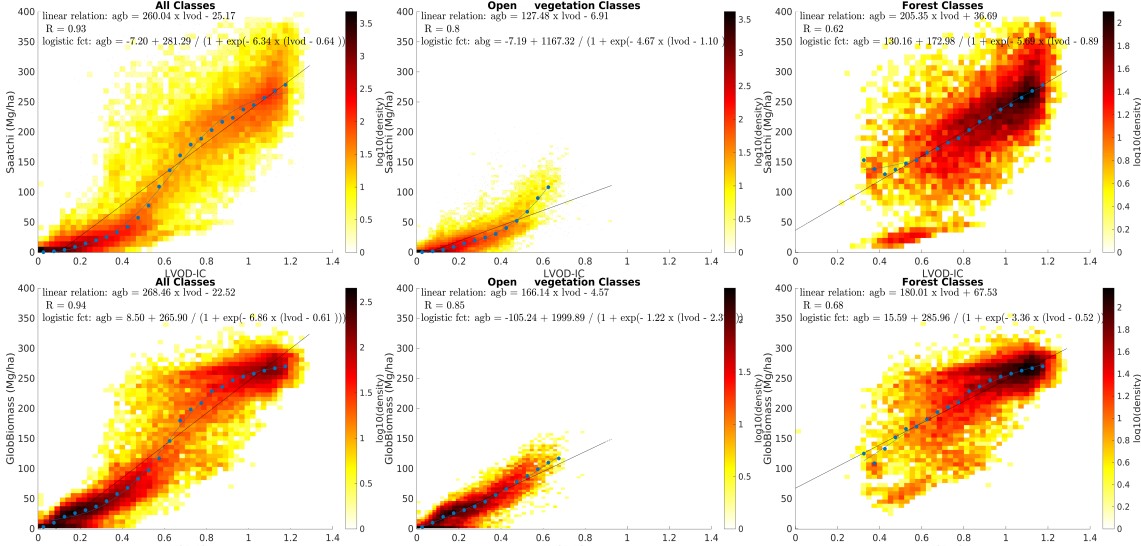

**Figure 4.** AGB Saatchi (**top row**) and GlobBiomass (**bottom row**) as a function of L-VOD over the tropical region. **left column** Figures are for all IGBP classes, **middle column** for low vegetation classes, and **right column** are for forest classes.

## 4. Discussion

The relationships L-VOD - AGB presented in this study suggest that the L-VOD depends on the density of the vegetation cover. For the less dense categories of vegetation, the relationship is slightly concave upward and is well described by a linear function. The spread of L-VOD may be explained by the fact that it is sensitive to vegetation water content and vegetation structure [27], as shown for croplands [36]. For dense forested areas, we observe a saturation of the L-VOD with AGB and a large spread of the L-VOD observations with respect to AGB especially for northern forests, which does not allow us to determine a clear relationship. However, all classes being considered, SMOS L-VOD presents a sigmoid shaped relationship with AGB estimates, confirming indications from a previous study over Africa [19].

### 4.1. Northern Latitudes

The relationships over the northern latitudes are not straightforward. SMOS L-VOD presents a seasonal dynamic due to the changing conditions between cold/warm or wet/dry that affect both the vegetation structures and the vegetation water content that varies with seasons. The strong agreement between L-VOD and the GlobBiomass AGB estimates are explained by the strong reliability of methods and observations used to estimate AGB in the northern forests by Santoro et al. [37]. Both C-band and L-band SAR backscatter contribute to the GlobBiomass AGB estimates of the northern forest, with an increase of the L-band component for increasing biomass. The density of forest is such that both types of backscatter used in the methodology of GlobBiomass, present sensitivity to the forest structural properties, even at the highest level of AGB. The models used to retrieve biomass fit thus well with the observations [38,39].

### 4.2. Tropical Areas

The whole tropical regions show more scatter relationship between the L-VOD and and the AGB estimates and specially for high AGB (>150 Mg/Ta). On the one hand the cluster of GlobBiomass AGB estimates (between 200 and 300 Mg/ha and L-VOD > 0.7, left column, bottom row Figure 4) is due to the rather constant value of AGB found in the wet tropics areas in the GlobBiomass map. This is a consequence of having available only a single observation of the L-band SAR backscatter to estimate AGB, and the necessity of having to apply a strong filter to reduce seams in the original SAR data. On the other hand, the AGB estimates from the Saatchi dataset spread over a wider range of values (150–400 Mg/ha, top row Figure 4). Although a single observation of the L-band SAR backscatter was used as well to derive the Saatchi map, the larger sensitivity to high AGB values is probably due to the fact that, contrary to the GlobBiomass dataset, AGB estimates from AGB in-situ measurements and from GLAS data were used and spatially extrapolated. The particular case of Africa is presented in Figure S3 and Table S1 in the Supplement as a comparison with previous work.

### 4.3. South America and Amazonia

The relationship L-VOD versus AGB was also studied in South America [18] using the AGB estimates by Reference [11]. Using SMOS level 2 data, Vittucci et al. [18] found a good relationship (coefficient correlation R of 0.71) when we found a correlation of 0.67 and 0.66 (versus Saatchi and GlobBiomass respectively) for this region and over forested lands (not presented in this study). The difference can be related to (i) SMOS datasets—level 2 (in Vittucci et al. [18]) and IC (in the present study) use the same IGBP land cover but with different parametrization [18,29], leading to slightly lower L-VOD with the level 2 than with IC ; (ii) different properties of the AGB estimates used as reference (i.e., Avitabile et al. [11] in Vittucci et al. [18]) which do not agree over dense forest and Amazonia [13].

## 5. Conclusions

This study evaluates the sensitivity of SMOS L-VOD to AGB estimates, extending previous studies over Africa and Amazonia to the whole globe. Compared to existing studies, this analysis takes advantage of two recent AGB maps to act as references to perform the evaluation. These datasets [12,33] are indeed coincident with SMOS acquisitions. The strong relationships between L-VOD and AGB found in a preliminary study for Africa [19] are confirmed at a global scale and in particular for open and low vegetation classes (open shrublands, woody savanna, grasslands, croplands). For the northern latitude forests the relationship is not as strong as for the tropics.

This paper is a first and necessary step for the development of a model to estimate an AGB from SMOS L-VOD. Specific work includes better selection of SMOS L-VOD according to the climate conditions (snow-free in the north latitudes, dry period in the Tropics, better description of waterbodies) and the development of a validation strategy with in-situ measurements representative of a SMOS pixel. SMOS L-VOD is strongly correlated with the AGB estimates especially with GlobBiomass [33] making the L-VOD an interesting dataset to study the evolution of northern latitudes ecosystems, and possibly considering a different relationship than the ones defined for the Tropical areas. Another development is to study the complementary of passive L-band microwave to other shorter wavelengths to better estimate the AGB especially over dense forests.

**Supplementary Materials:** The following are available online at http://www.mdpi.com/2072-4292/12/9/1450/s1, Figure S1: L-VOD versus AGB at global scale ; AGB from Saatchi et al. is presented on the left panel side and AGB from GlobBiomass et al. on the right panel. IGBP classes belonging to the low vegetation group are shown here, Figure S2: same as Figure S1 for the IGBP classes belonging to the forest group, Figure S3: AGB Saatchi (top row) and GlobBiomass (bottom row) as a function of L-VOD over Africa. left column Figures are for all IGBP classes, middle column for low vegetation classes, and right column are for forest classes. On the left column the relation found by Rodríguez-Fernández et al, Table S1: Statistics over Africa.

**Author Contributions:** A.M., N.J.R.-F, Y.H.K. designed and planned the research; A.M., N.J.R.-F., E.B., performed the data processing; M.S., S.S. provided the AGB datasets; M.S., S.S., S.M. provided expertise on AGB estimations and radar datasets; All authors participated in the writing and provided comments and suggestions. All authors have read and agreed to the published version of the manuscript.

**Funding:** This research was funded by CNES (Centre National d'Etudes Spatiales) TOSCA program, by the CATDS and by the ESA contract No. 4000117645/16/NL/SW Support To Science Element SMOS+Vegetation

**Acknowledgments:** A.M., N.J.R.-F., E.B., Y.H.K. acknowledge support by the ESA contract No. 4000117645/16/NL/SW Support To Science Element SMOS+Vegetation and by CNES (Centre National d'Etudes Spatiales) TOSCA program. SMOS IC were obtained from the "Centre Aval de Traitement des Données SMOS" (CATDS), operated for the CNES (France) by IFREMER (Brest, France). The authors would like to thank the European Space Agency (ESA) Support to Science Element (STSE) programme and SMOS Export Support Laboratory (ESL) for funding this study.

**Conflicts of Interest:** The authors declare no conflict of interest.

## Abbreviations

The following abbreviations are used in this manuscript:

| | |
|---|---|
| AGB | Above Ground Biomass |
| L-MEB | L-band Microwave Emission of the Biosphere |
| SMOS | Soil Moisture and Ocean Salinity |
| TB | Brightess Temperature |
| L-VOD | L-band Vegetation Optical Depth |

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
