# Peer review of "Evaluation of the Sensitivity of SMOS L-VOD to Forest Above-Ground Biomass at Global Scale"

_remotesensing, doi:10.3390/rs12091450_

Round 1

Reviewer 1 Report

This paper shows the ability of SMOS- L-VOD (with 10 years of data) to “follow” AGB from two database. The authors presented new finding in different environments, not previously studied. This study thoroughly covered most biomes described in the literature, although the statistic used is very basic and should be improved. In my opinion, this  paper should be shorten to a letter.

I have few comments and suggestions.

1-Authors say “Note that we do not attempt to estimate AGB from the L-VOD in this study, so only the correlation coefficient is computed” in line 148. If so, why did you present two regression models?

There are different statistics that describe the relationship between two variables that can be used here to shorten the paper. You used a linear regression (fig 3 and 4) showing only results of R but it can be done in a table. Figure 3 and 4 can be deleted.

2- Figure 3 and 4 display linear relationship but then in Fig 5 and 6, you used logistic function (as Rodríguez-Fernández et al did). How linear relationships ended in no linear relationships? Is there an explanation to add in the paper?

3- You provide R values for linear and logistic functions for different classes, but your conclusion is that the relationship is good…. This, very useful conclusion can be done with a simpler analysis, in a letter, adding more statistics. If your ongoing research is to estimate AGB from L-VOD, using these regression functions, you should mention it.

 4- Results from Africa showed in Discussion, should be moved to results, and then discussed (line220).

5-In line 290, you concluded that “Secondly, this study also shows that the relationship found for Africa is applicable to the whole tropics”.

How a “good relationship” can be applicable to another place, without a model?

Then, in 293 you said “making the L-VOD an interesting dataset to study the evolution of northern latitudes ecosystems, considering a different relationship than the ones defined for the Tropical areas.”

It sounds to me that you are looking for something else than the agreement between L-VOD and AGB, it is not clear whether you want to show a “statistic model” or what.

6- I noted you did not used well enough either commas or semicolons.

Author Response

Dear,

We would like to thank the reviewer for his/her comments. We agree with the suggestion to shorten the manuscript. The new version has been adapted to the letter format. Please find our responses to your comments below, in italic, after your comments.

"This paper shows the ability of SMOS- L-VOD (with 10 years of data) to “follow” AGB from two database. The authors presented new finding in different environments, not previously studied. This study thoroughly covered most biomes described in the literature, although the statistic used is very basic and should be improved. In my opinion, this  paper should be shorten to a letter."

Note that some figures of the previous version are now part of the supplementary document. Figures 1 and 2 of the previous version have been combined into figure 1.

"I have few comments and suggestions.

1-Authors say “Note that we do not attempt to estimate AGB from the L-VOD in this study, so only the correlation coefficient is computed” in line 148. If so, why did you present two regression models?"

We chose to depict 2 functions because a linear function seemed appropriate for the comparison per biomes. When the individual classes were grouped, the linear fit was less obvious. We kept the linear fit as it allows us to compute the correlation coefficicent. This coefficient was helpful to discuss the sensitivity of our LVOD with respect to the AGB dataset. Rodriguez et al. used both the Pearson’s correlation coefficient (linear fct) and the Spearman's rank correlation coefficient for monotonic relationships. The differences were very small so we decided to select the Pearson’s R as a linear function seemed appropriate for the comparison per biomes (Supplementary material).

"There are different statistics that describe the relationship between two variables that can be used here to shorten the paper. You used a linear regression (fig 3 and 4) showing only results of R but it can be done in a table. Figure 3 and 4 can be deleted."

Previous figures 3 and 4 (comparison per classes) were put in a supplementary material document with the comments related to the figures.

We also moved the part on Africa in the supplementary material document as it is somehow covered by Rodriguez et al. Paper, eventhough we present results with more recent and updated AGB maps.

"2- Figure 3 and 4 display linear relationship but then in Fig 5 and 6, you used logistic function (as Rodríguez-Fernández et al did). How linear relationships ended in no linear relationships? Is there an explanation to add in the paper?"

Figures 3 and 4 were specific for each biomes. The distribution of points for each biomass suggest to use a linear function as a fit. Each of the individual cases (per class) cover different ranges of VOD and AGB. Once concatenated by groups, it does not imply that the distribution of points describe a linear trend. A function can be monotuous and linear by part.

"3- You provide R values for linear and logistic functions for different classes, but your conclusion is that the relationship is good…. This, very useful conclusion can be done with a simpler analysis, in a letter, adding more statistics. If your ongoing research is to estimate AGB from L-VOD, using these regression functions, you should mention it."

A sentence was added to clearly state that our goal is to be able to estimate an AGB from the L-VOD but this is beyond the scope of this paper. This first step consists in evaluating the L VOD, to check if it is related to AGB for all biomes and regions of the Earth surface. We simply use the coefficient of correlation as an indicator and to support our comments. To perform other statistics (rmse, bias for instance) we would need to estimate an AGB from the L-VOD and a model. Then one could compare the data. We thought it was not the objective of this evaluation.

" 4- Results from Africa showed in Discussion, should be moved to results, and then discussed (line220)."

Figure 8 on Africa is moved to the supplementary material, as well as the comments on the Africa region. It was necessary to shorten the manuscript.

"5-In line 290, you concluded that “Secondly, this study also shows that the relationship found for Africa is applicable to the whole tropics”.

How a “good relationship” can be applicable to another place, without a model?"

We want to clarify the message of this paper. It is correct to assume that we aim at finding a model to estimate an AGB from the L-VOD. However, this step requires more work such as a better selection of L-VOD according to climate conditions. The validation of such an estimate is a difficult task as it requires in-situ measurements representative at SMOS pixel.

By this study we wanted to evaluate the possibility of L-VOD to monitor AGB everywhere. This is the first step towards a more detailed approach to be able to propose an appropriate model.

However, we agree with the comment and should have specified more clearly our intention. We added some sentences in the conclusion of the new manuscript.

«This paper is a first and necessary step for the development of a model to estimate an AGB from SMOS L-VOD. Specific work includes better selection of SMOS L-VOD according to the climate conditions (snow-free in the north latitudes, dry period in the Tropics, better description of waterbodies) and the development of a validation strategy with in-situ measurements representative of a SMOS pixel. » 

" Then, in 293 you said “making the L-VOD an interesting dataset to study the evolution of northern latitudes ecosystems, considering a different relationship than the ones defined for the Tropical areas.”

It sounds to me that you are looking for something else than the agreement between L-VOD and AGB, it is not clear whether you want to show a “statistic model” or what."

The reviewer is right. The next step is to be able to estimate the AGB from L-VOD using a model. We thought that we need first to present the sensitivity of L-VOD to AGB at global scale as it has not been done. The letter format is perfect for that purpose (i.e. one idea/objective)

We added a sentence to support this comment in the conclusion. This paper is a first and necessary step for the development of a model to estimate an AGB from SMOS L-VOD.

« Specific work includes better selection of SMOS L-VOD according to the climate conditions (snow-free in the north latitudes, dry period in the Tropics, better description of waterbodies) and the development of a validation strategy with in-situ measurements representative of a SMOS pixel. » 

" 6- I noted you did not used well enough either commas or semicolons."

The text was read and corrected by an english speaker.

Reviewer 2 Report

The authors present the sensitivity analysis between the Optical depth and AGB from two recent published dataset showing a moderate linear correlation. Overall, the paper is written and organized very well and discussion looks reasonable. 

There are some minors that need to be revised. 

  1. present the AGB estimation accuracy from Saatchi and GlobBiosmass in Section 2.2 and 2.3. 
  2. Spell out the IC in line 252.

Author Response

Dear reviewer.

Thanks for your comments. Note that considering the other reviewers comments, the manuscript has been shortened and is now presented as a letter. It does not change the main objective of the manuscript but it is more to the point.

Please find below your responses to your comments.

Regards

There are some minors that need to be revised. 

    1.  

present the AGB estimation accuracy from Saatchi and GlobBiosmass in Section 2.2 and 2.3. 

It is somewhat difficult to have one number that quantifies the accuracy of a global product because the quality of the map differs depending on 1) reference data (not the same quality everywhere) and 2) AGB retrieval (not the same performance everywhere). For Globbiomass, RMSD between 50% and 90% was obtained, depending on the biome (tropical, subtropical, temperate and boreal) and a global RMSD of 92%. But this RMSD differs depending on the AGB level.

IC was already described line 57 (new version of the manuscript)

Reviewer 3 Report

The paper presented a very interesting analysis and focused on a highly important topic. A few specific comments are below-

1) I felt like the paper requires improvement in terms of sentence structure and style, especially in the introduction. Many of the sentences are convoluted and ultimately, not providing the actual message. For example, lines 19-21 ( in the context of....), line 25 (but it is a....).

2) Many of the acronyms are not described, e.g line 28 - BIOMASS, NiSAR. 

3) line 33: being insensitive to...?

4) line 34: VOD in the bracket?

5) Also, some of the acronyms were described in the bracket, e.g. line 2  AGB (Above Ground Biomass)..shouldn't it be otherwise?

6) The method section could be explained better, maybe a flow chart explaining the overall data analysis steps?

7) Can the sections 2.1-2.4 be grouped under one section and then each of them could be sub-headings? The way it is organized now reads a little confusing and initially, I was thinking why these are not part of the introduction and then, I realized these are some of the input data to the model.  

8) In the results, the graphs are showing 'Saatchi' while in the description and tables, it is 'Saatchi15'. Can these be consistent?

9) Line 291: What type of specific analysis? Providing some examples would be nice for the readers.

Author Response

Dear Reviewer

Thank you for your comments. Please find our responses in italic below each of your comments

Best regards

" The paper presented a very interesting analysis and focused on a highly important topic. A few specific comments are below-"

We thank the reviewer for her/his time and comments to help us to improve our manuscript. Please note that, considering another review’s comment, the paper was shortened to fit the format of a letter with the main objective to evaluate the sensibility of the L-VOD to AGB estimates at global scale. We had to move some parts to a « supplementary material » document.

"1) I felt like the paper requires improvement in terms of sentence structure and style, especially in the introduction. Many of the sentences are convoluted and ultimately, not providing the actual message. For example, lines 19-21 ( in the context of....), line 25 (but it is a....)."

Some sentences of the introduction have been re written and the manuscript was shortened, hopefully it helps to clarify our text

"2) Many of the acronyms are not described, e.g line 28 - BIOMASS, NiSAR. "

Biomass is not an acronym. It stands for Biomass. We agree though with NISAR which has been detailed «  NASA-ISRO Synthetic Aperture Radar (NISAR) » lines 27-28 (Introduction).

"3) line 33: being insensitive to...?"

You are right, it is corrected.

"4) line 34: VOD in the bracket?"

it is corrected.

"5) Also, some of the acronyms were described in the bracket, e.g. line 2  AGB (Above Ground Biomass)..shouldn't it be otherwise?"

Correct, it has been updated.

"6) The method section could be explained better, maybe a flow chart explaining the overall data analysis steps?"

A flow chart is a good idea. However, we were recommended to shorten the manuscript which is not compatible with adding a new figure. The text was modified to hopefully clarify the method.

"7) Can the sections 2.1-2.4 be grouped under one section and then each of them could be sub-headings? The way it is organized now reads a little confusing and initially, I was thinking why these are not part of the introduction and then, I realized these are some of the input data to the model.  "

These suggestions have been considered and this section is now divided into a « data » part and a « method » part

"8) In the results, the graphs are showing 'Saatchi' while in the description and tables, it is 'Saatchi15'. Can these be consistent?"

Yes. We thought that it was confusing so we kept « Saatchi » instead.

" 9) Line 291: What type of specific analysis? Providing some examples would be nice for the readers."

High latitudes are complex because of the presence of snow, large open water bodies in Summer. Anyway, we agree with the comment and specified our statement in the conclusion

« Specific work includes better selection of SMOS L-VOD according to the climate conditions (snow-free in the north latitudes, dry period in the Tropics, better description of waterbodies) and the development of a validation strategy with {\it in-situ} measurements representative of a SMOS pixel.»

Round 2

Reviewer 1 Report

I would like to thank the authors for considering my opinion and answering my questions. The revised manuscript is a very stimulating reading for RS readers.

I had noted in Figure 2 and 4 AGB, there is a kind of saturation at 300 Mg/ha. What can cause this? If there is a simple explanation you may considering wrtitting it.

Best Regards

Author Response

Dear Reviewer

Thank you for your reply.  In response to  your comment :" I had noted in Figure 2 and 4 AGB, there is a kind of saturation at 300 Mg/ha. What can cause this? If there is a simple explanation you may considering wrtitting it."

Indeed there is a saturation effect observed for high AGB estimates, especially in the GlobBiomass dataset. This "saturation" is an effect of having a very small
range of estimated biomass in the dense wet tropics in the GlobBiomass
dataset.

Two remarks in the paper mention this issue and provide the reader with the explanations of this effect.

*) one line 83, Page 3 describing the GlobBiomass dataset : "Nonetheless, the estimates are affected by an increasing underestimation in wet tropical and temperate forests when AGB is above 250 Mg/ha."

and one in the discussion

*) page 7, ine 172 : " On the one hand the cluster of GlobBiomass AGB
estimates (between 200 and 300 Mg/ha and L-VOD > 0.7, left column, bottom row Figure 4) is due to the rather constant value of AGB found in the wet tropics areas in the GlobBiomass map. This is a consequence of having available only a single observation of the L-band SAR backscatter to estimate AGB, and the necessity of having to apply a strong filter to reduce seams in the original SAR data"

Regards,